ecology, environmental science, plant science

chlorophyll fluorescence, *Dicranum scoparium*, $\delta^{13}C$, $\delta^{18}O$, Polytrichales, *Sphagnum* spp.

**Author for correspondence:**
Jessica Royles
e-mail: jr328@cam.ac.uk

†Present address: Lancaster Environment Centre, Lancaster University, Lancaster, LA1 4YQ, UK.

# Stable isotope signals provide seasonal climatic markers for moss functional groups

Jessica Royles, Sophie Young† and Howard Griffiths

Department of Plant Sciences, University of Cambridge, Downing Street, Cambridge CB2 3EA, UK

JR, 0000-0003-0489-6863

Living moss biomass and archival peat deposits represent key indicators of present and past climatic conditions, but prediction of future climatic impacts requires appropriate marker species to be characterized under a range of contemporary conditions. Stable isotope signals in high latitude moss deposits offer potential climatic proxies. Seasonal changes in $\delta^{13}C$ and $\delta^{18}O$ of organic material (cellulose) in representative functional groups, and associated photosynthetic activity (as chlorophyll fluorescence) have been compared across East Anglia, UK, as a function of tissue water content. Representative species from contrasting acid bog, heathland, and fen woodland habitats were selected for monthly sampling of recent growth tissues between spring 2017 and autumn 2018, with isotopic signals in purified cellulose compared with tissue water, precipitation, and nearby groundwater signals. Sphagnum and Polytrichum groups, which tend to dominate peat formation, provided contrasting and complementary indicators of seasonal variations in carbon assimilation. Cellulose $\delta^{18}O$ signals from *Sphagnum* spp. demonstrate seasonal variations in source precipitation inputs; carbon isotope signals in *Polytrichum* spp. indicate evaporative demand and photosynthetic limitation.

## 1. Introduction

A better understanding of the timing and extent of active moss metabolism is crucial for the interpretation of peatland palaeoclimate archives [1], for the improved accuracy of global vegetation models [2], and for predicting ecosystem responses to climate change [3]. The tight coupling of poikilohydric mosses to the environment makes them useful tools to probe relationships between plants and climate [4]. In order to interpret the high resolution stable isotope data preserved over thousands of years in both Antarctic moss banks (e.g. [5,6]) and northern peatlands (e.g. [7,8]), it is crucial to identify, for representative marker species, how preserved proxies are coupled with the environmental conditions when mosses are physiologically active.

Unlike higher plants, bryophyte gametophytes lack both a typical vascular conducting system for water and stomatal pores for control of transpiration. Photosynthesis can only occur when moss tissue is hydrated. However, external capillary water acts as a diffusion barrier for carbon dioxide, and thus, when the moss is saturated the rate of photosynthesis will be lower than the rate just prior to desiccation-induced metabolic shut down [9,10]. As a consequence of this paradox, water management strategies vary between moss life forms [11]. Growing in wet environments, *Sphagnum* mosses maintain a high, constant, water content, with maximum assimilation rates occurring at 70–100 times dry weight [12]. At the other extreme, highly desiccation-tolerant species such as *Syntrichia ruralis* [13] grow in exposed habitats and go through numerous desiccation-rehydration cycles [10]. Polytrichale mosses are ectohydric with central hydroids for water conduction that deliver water under tension to

photosynthetic tissues [14]. Thus even mosses that co-occur, such as *Sphagnum* and *Polytrichum* spp., may be metabolically active at different times and respond differently to the prevailing environmental conditions.

The $^{13}C/^{12}C$ ratio of cellulose ($\delta^{13}C_C$) is dependent upon both the isotopic composition of atmospheric $CO_2$ ($\delta^{13}C_a$) and the extent of discrimination against $^{13}CO_2$ during photosynthesis [15], integrated over the lifetime of the plant. Under wet conditions, the diffusion into the chloroplast is limited and discrimination against $^{13}CO_2$ is lower [9,16]. During dry periods the external capillary water evaporates, and diffusion limitation is relieved, so increasing discrimination is expressed against $^{13}CO_2$, until desiccation-induced metabolic shut down occurs. Thus, $\delta^{13}C_C$ of contemporary *Sphagnum* samples reflects the wetness of the growing conditions [1], while in more desiccation-tolerant species it reflects the optimality of the conditions for photosynthesis over the growth period [4]. The resultant isotope signal integrates long periods of low discrimination during reduced assimilation rates when tissue surface moisture is plentiful, interspersed with brief periods of higher discrimination during high assimilation rates immediately prior to desiccation.

The $^{18}O/^{16}O$ ratio of cellulose ($\delta^{18}O_C$) depends on the isotopic composition of the source water, any isotopic changes undergone by the source water prior to incorporation into cellulose, and the fractionations associated with the synthesis of cellulose, approximately 27‰ [17]. For desiccation-tolerant mosses, tissue water is likely to undergo evaporative enrichment prior to incorporation into cellulose: there is also evidence that assimilation ($\delta^{13}C_C$ depleted during desiccation) and growth ($\delta^{18}O_C$ formed during rehydration) can be temporally separated, and thus the relationship between the two measured isotope values in a single sample may be uncoupled [10]. By contrast, for *Sphagnum* mosses, with a relatively constant, high water content, $\delta^{13}C_C$ and $\delta^{18}O_C$ may be negatively correlated due to the simultaneous impacts of the surface water diffusion limitation ($\delta^{13}C_C$ more enriched) and lower evaporative enrichment ($\delta^{18}O_C$ more depleted) [1].

Peat accumulation is a critical component of global carbon storage [18] and understanding the physiological control of contributory moss growth is critical for predicting how climate change will impact continued carbon sequestration across boreal regions [4]. Ground truthing of proxies for photosynthetic activity in moss, and hence peatlands, allows measurements to be scaled forward for climatic projections, or applied retrospectively to account for shifts in palaeohistorical peat bank records and past climate [1,19]. Chlorophyll fluorescence measurements provide an instantaneous assessment of moss vitality, and can be related on a larger spatial scale to light-induced fluorescence transient (LIFT; [20,21]) and solar-induced fluorescence (SIF; [22–24]) measurements. The stable isotope ($\delta^{13}C$ and $\delta^{18}O$) compositions of organic matter are integrators of seasonal shifts in assimilatory conditions, water sources and water availability that can be preserved over thousands of years [4,25,26]. In order to address the complex interactions between environmental physiology and stable isotope composition, we tracked the real-time variations in representative moss species over two growing seasons. The objective was to characterize how $\delta^{13}C$ and $\delta^{18}O$ signals represent markers of photosynthetic carbon assimilation and hydraulic status, respectively, for key moss functional groups. We show that stable isotope markers in organic material, in conjunction with chlorophyll fluorescence, provide contrasting biomarkers for two key moss

orders (Sphagnales and Polytrichales) which could be used comparatively to assess past and future climatic impacts.

# 2. Methods

## (a) Meteorological data

Monthly meteorological records from Cambridge were obtained from the Computing Laboratory, University of Cambridge (52.2108° N, 0.0914° E [27]), while moss surface level temperature and relative humidity were recorded hourly by a TinyTag2 datalogger (Gemini Data loggers, UK) embedded in a moss turf under partial shading at the Cambridge University Botanic Garden to provide measurements comparable with the meteorological data, as installations were not permitted at the sampling locations (electronic supplementary material, figures S1 and S2).

## (b) Fieldwork

Fieldwork was carried out approximately monthly from April 2017 until September 2018 at three field sites across East Anglia (Dersingham Bog, Brandon Country Park, and Wicken Fen; electronic supplementary material, figure S1 and table S1) that incorporated bog, heathland, shaded woodland, and fen habitats. Target species were identified that represented a range of moss life forms, growing under different conditions, which also provided comparable analogues for moss species used in palaeoclimate work. The ectohydric Polytrichales, present in Antarctic peat bank palaeoclimate archives were represented by *Polytrichum commune* (Dersingham) and *Polytrichastrum formosum* (Brandon). *Dicranum scoparium*, a Dicranale and similar to the Antarctic peat bank species *Chorisodontium aciphyllum*, was also present at Dersingham and Brandon. Four *Sphagnum* spp. species, characteristic of wet environments were sampled (Dersingham). Finally, four Hypnale (*Pseudoscleropodium purum*, *Pleurozium schreberi*, *Brachythecium rutabulum*, *Calliergonella cuspidate*, *Kindbergia praelonga*) and one Bryale species (*Aulacomnium palustre*) were sampled from across the sites. Voucher specimens were collected and their identity verified by experienced bryologists (electronic supplementary material, table S1, CD Preston, MO Hill). Permission was gained from all the landowners/managers prior to the commencement of fieldwork.

Each visit, three surface samples approximately 4 cm$^2$ in area and 2 cm deep, incorporating the growing tips, were harvested from each moss species, at each site it was present. From June 2017 onwards, field chlorophyll fluorescence measurements were made at the moss growing tips (Walz MINI-PAM II Walz, Effeltrich, Germany) with three measurements ($F'$, $F_m'$) taken using the leaf clip before harvest on separate sub-samples under ambient light (photosynthetically active radiation (PAR)), which was measured by the light sensor. Following harvest, moss samples were stored in sealed plastic bags, and transported to the laboratory. After at least 30 min of dark adaption, three further fluorescence measurements were made in the dark ($F$, $F_m$). Fluorescence yield was calculated in the dark ($F_m - F)/F_m$) and light (($F_m' - F')/F_m'$). Electron transport rate (ETR) was calculated as: $((F_m' - F')/F_m') \times PAR \times 0.42$, where 0.42 is the product of light absorptance by an average green leaf (0.84) times the fraction of absorbed quanta available for photosystem II (0.5). Approximately 0.3 g of each sample was weighed and dried to a constant mass at 70°C to establish field relative water content (RWC = (Fresh weight − Dry weight)/Dry weight)). The growing tips of the dry moss were transferred to Soxhlet thimbles for cellulose extraction following standard procedures [28]. Local water samples were collected in 10 ml vials from sitting bog water (Dersingham), a lake (Brandon), and a lode (Wicken) on each visit. Precipitation was collected in a rain gauge near the centre of the field site area (Ely, 52.40° N,

0.26° E, 36 km from Dersingham, 26 km from Brandon, 22 km from Wicken), throughout the sampling period following rain events. A sub-sample of a selection of fresh field mosses harvested in July 2017 underwent enzymatic photo-spectrophotometric sucrose content analysis following standard procedures [29].

## (c) Isotope analysis

Fresh water samples underwent $^{18}O/^{16}O$ isotope analysis at Lancaster Environment Centre. $\delta^{18}O$ analysis was completed by pyrolysis on a VarioPyrocubeEA and the $\delta^{18}O$ were measured on an Isoprime100 isotope ratio mass spectrometer (IRMS) with a precision of ±0.5‰.

$\delta^{18}O_c$ and $\delta^{13}C_C$ analysis was undertaken by EA-IRMS at the Godwin Laboratory for Palaeoclimate Research, University of Cambridge. $\delta^{18}O_c$ was measured using a Thermo Finnigan TC/EA attached to a Thermo Delta V mass spectrometer via a ConFlo 3 with an analytical precision better than 0.4‰. $\delta^{13}C_C$ was analysed using a Costech Elemental Analyser attached to a Thermo DELTA V mass spectrometer in continuous flow mode. The precision of analyses was better than 0.1‰.

## (d) Statistical analysis

Statistical analysis was carried out in R (v. 4.0.5) [30]. The significance of the different proportions of contrasting moss groups with low dark yield measurements was tested using the $\chi^2$-test. As they were not normally distributed the difference in sucrose content between *Sphagnum* and non-*Sphagnum* was tested using the Wilcoxon signed-rank test. Using the lme4 package [31] a mixed-effect model was fitted to the water and $\delta^{18}O_c$ data with *post hoc* pairwise testing using Tukey tests and the Kenward-Rogers degrees of freedom model. The rmcorr package [32] was used to calculate the relationship between RWC and $\delta^{18}O_C$ (figure 3*d*), taking account of the repeated measurements.

# 3. Results

## (a) Meteorological data

Throughout 2017 and 2018, mean monthly air temperatures ranged from 2 to 19°C, with a minimum of −7.6°C and maximum of 31.7°C (electronic supplementary material, figure S2a). Monthly rainfall ranged from a trace to 100 mm, while sunshine hours peaked at over 300 h. Moss surface temperatures followed air temperature patterns but both the maximum (43°C) and the minimum temperatures (−2°C) were higher (electronic supplementary material, figure S2b).

## (b) Fresh moss analysis

During the 18 fieldwork sessions 316 fresh moss samples were collected and separated into four life form categories: Sphagnales ($n = 90$), Polytrichales ($n = 36$), Dicranales ($n = 35$), and the remainder, a group of Hypnale and Bryale species ($n = 155$). Field RWCs ranged from 0 to 27 times the dry weight. *Sphagnum* mosses were always the wettest, with a RWC around 12 (figure 1*a*). *Sphagnum* mosses were driest in June 2017 (month 6), which coincided with the lowest photosynthetic efficiency, measured as fluorescence yield in the field, of approximately 0.25 (figure 1*b*). While Polytrichales tended to be the highest, patterns of field yield were similar across the mosses, increasing from the minimum in the summer of 2017, towards maximal values in winter.

Partially reflecting degree of exposure and incident light levels, ETR was generally highest in the *Sphagnum* mosses,

and highest during summer months (figure 1*c*). The Dicranales had the lowest ETR, which was particularly apparent in summer 2018, when peaks in ETR were measured in all other mosses. Field yield ($F_v'/F_m'$) is an indication of *in situ* photosynthetic light use ability and 72% of samples had yield values greater than 0.5 (figure 1*d*) compared to 83% of dark yield ($F_v/F_m$) values, showing that photosynthetic competence could recover following dark acclimation, except in the very driest of moss tissues (figure 1*e*). Only 7% of Sphagnale dark yield readings were less than 0.5, and three of these were during winter months (electronic supplementary material, table S3). A higher proportion of low dark yield measurements were found in the Hypnales, Bryales, and Dicranale samples than Polytrichales and Sphagnales ($\chi^2$ (1, $N = 322$) = 5.27, $p = 0.022$), with 29% of dark yield measurements in the Dicranales less than 0.5, and low readings occurring in the same months at the Brandon and Dersingham sites (electronic supplementary material, table S3). Sucrose concentration, measured as an indicator of both photosynthetic activity and an osmotic component, was significantly higher, by approximately 10-fold, in non-*Sphagnum* mosses than *Sphagnum* mosses collected in the field ($W = 135$, $p < 0.01$; figure 1*f*).

## (c) $\delta^{18}O$ composition of source waters and cellulose

The $\delta^{18}O$ composition measured in rainfall in East Anglia (figure 2*a*) had a mean of −5.41‰. The rainfall was most depleted early in 2018 at −8‰ and most enriched during the summer of 2017 at −2‰. More depleted samples were collected as snow, down to −15‰ (figure 2*a*). Local fresh water samples collected from surficial pools at Dersingham Bog had an isotopic composition indistinguishable from rain (offset = 0.20‰, $p = 0.94$), but the groundwater sources were significantly enriched compared to rain at Wicken Fen (offset = 2.18‰, $p < 0.001$), and Brandon (offset = 4.2‰, $p < 0.001$). The $\delta^{18}O_C$ values of non-*Sphagnum* samples were consistently around 4‰ enriched compared to the contemporaneous *Sphagnum* samples, and were more variable on a month-to-month basis. An exceptional increase in Sphagnum $\delta^{18}O_C$ in late spring of 2017 (mirrored in the rain and cellulose signals of other moss species) was associated with a period of extended sunshine hours and high surface temperatures (electronic supplementary material, figure S2a). For Sphagnum samples, all from Dersingham Bog and reflecting the similar water compositions, the offset between $\delta^{18}O_C$ and both rain and local water ($\delta^{18}O_c$–$\delta^{18}O_w$) was very similar with a median of 27.5‰ (figure 2*b*), suggesting little evaporative enrichment relative to source water occurred during new growth. For the non-Sphagnum samples, combined from all three sites, the higher variation in $\delta^{18}O_C$ and increased degree of $\delta^{18}O_w$ enrichment in the local groundwater was also associated with a higher offset between cellulose and water ($\delta^{18}O_c$–$\delta^{18}O_w$) (29‰ for local water, 31‰ for precipitation, figure 2*b*), suggesting a higher degree of evaporative enrichment in these more exposed tissues during metabolism and growth.

## (d) $\delta^{13}C$ and $\delta^{18}O$ composition of cellulose

There were similar, marked seasonal variations in $\delta^{13}C_c$ across the three non-Sphagnum moss groups, starting around −27.5‰ in spring 2017, before becoming substantially more depleted, approaching −30‰ by September 2017 (figure 3*a*). $\delta^{13}C_c$ then tended to become more enriched

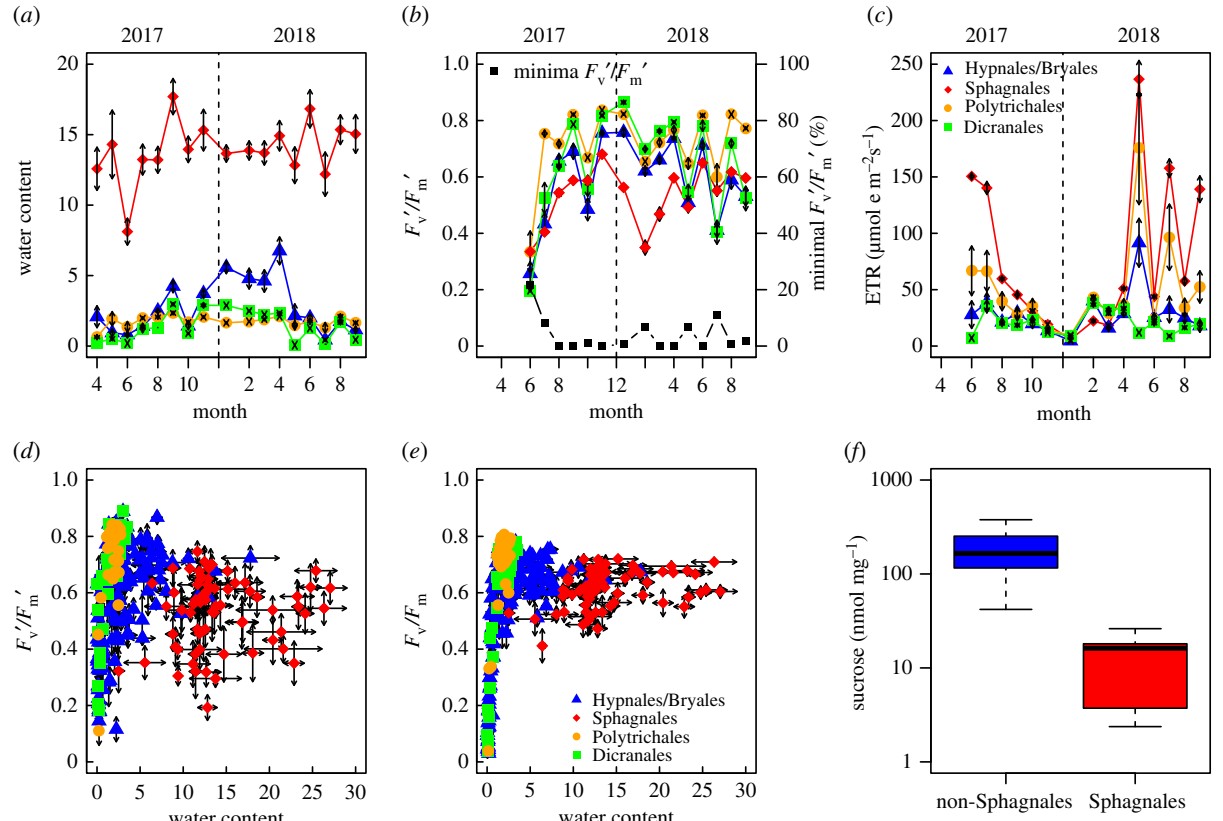

**Figure 1.** Analysis of all fresh field moss samples, divided into Sphagnales (red diamonds), Polytrichales (orange circles), Dicranales (green squares), and Hypnales/ Bryales (blue triangles). (*a*) Mean (±s.e.) water content (WC = (FW − DW)/DW) of moss samples per month, (*b*) mean (±s.e.) chlorophyll fluorescence yield as $F_v'/F_m'$, also proportion of individual measurements ($n = 162$) in which no yield was detectable, and (*c*) mean electron transport rate (ETR) measured each field trip under field conditions over time; mean (±s.e., $n = 3$) chlorophyll fluorescence yield measured (*d*) in the field ($F_v'/F_m'$) and (*e*) after at least 30 min dark adaptation ($F_v/F_m$) for all samples as a function of water content (mean ± s.e., $n = 3$). (*f*) Mean (±s.e., $n = 3$) sucrose concentration of mosses collected from the field (NS: $n = 7$, S: $n = 5$), NB logarithmic scale on *y*-axis. NS = non-Sphagnum, S = Sphagnum. (Online version in colour.)

over the winter and spring, up to approximately −27.5‰, before declining to −29‰. There was little seasonal variation in the $\delta^{13}C$ values for the *Sphagnum* mosses, with most measurements between −27‰ and −28‰ (figure 3*a*). There were no clear relationships for any of the moss groups between the $\delta^{13}C_c$ and $\delta^{18}O_c$ values (figure 3*b*), nor between $\delta^{13}C_c$ and water content at the time of collection (figure 3*c*). $\delta^{18}O_C$ ranged from 19‰ to 31‰ and was significantly negatively correlated with water content at the time of sampling (figure 3*d*, repeated measures correlation $r = −0.364$, d.f. = 297, $p < 0.001$). The generally wetter *Sphagnum* samples tended to be at the lower end of the $\delta^{18}O_c$ range (electronic supplementary material, figure S3), mostly 18–24‰, while the non-*Sphagnum* mosses had more enriched values, mostly from 23 to 27‰.

The Polytrichale $\delta^{13}C_c$ values fell into two distinct groups (figure 3*e*): those more negative than −28‰ were *P. formosum* samples from the dry, sandy Brandon Country Park, while those less negative than −28‰ were *P. commune* samples from the wet Dersingham Bog. *Dicranum scoparium* was also sampled at both Brandon and Dersingham, and there was a clear difference in $\delta^{13}C_c$ values between sites (figure 3*g*): all those more negative than −28.5‰ were from the drier Brandon, while those less negative than −28‰ were from the wetter Dersingham. In neither the Polytrichales (figure 3*f*) nor the Dicranales (figure 3*h*) was there a comparable division in the $\delta^{18}O$ values, with samples from both sites ranging from 20 to 30‰.

# 4. Discussion

Using chlorophyll fluorescence and stable isotope measures we provide a detailed analysis of contemporary *in situ* moss photosynthetic activity over two growing seasons across East Anglia, UK. We show that two genera with contrasting growth forms, which often coexist in Boreal habitats, provide parallel and comparative proxies for climatic signals for a given habitat. The $^{18}O/^{16}O$ signal in cellulose from fully hydrated *Sphagnum* spp. provides a key marker of precipitation inputs, while the $^{13}C/^{12}C$ signal in *Polytrichale* spp. is consistent with photosynthetic carbon assimilation activity, as moderated by evaporative demand and extent of rehydration.

## (a) Contrasting mire, fen, and heathland provided diverse moss functional group habitats

The high water storage capacity of *Sphagnum* spp. provides a relatively stable environment to support photosynthesis: providing a consistent and continuous support for photosynthesis and the integration of hydraulic signals into accumulating carbon. By contrast, the Dicranales, Hypnale, and Bryale mosses, growing in heathland, fen, and woodland, had a wide range of water contents due to their rapid and dynamic coupling to their immediate surroundings. Polytrichales, which, with a more advanced vascular function [14] are the mosses most isolated from their environment,

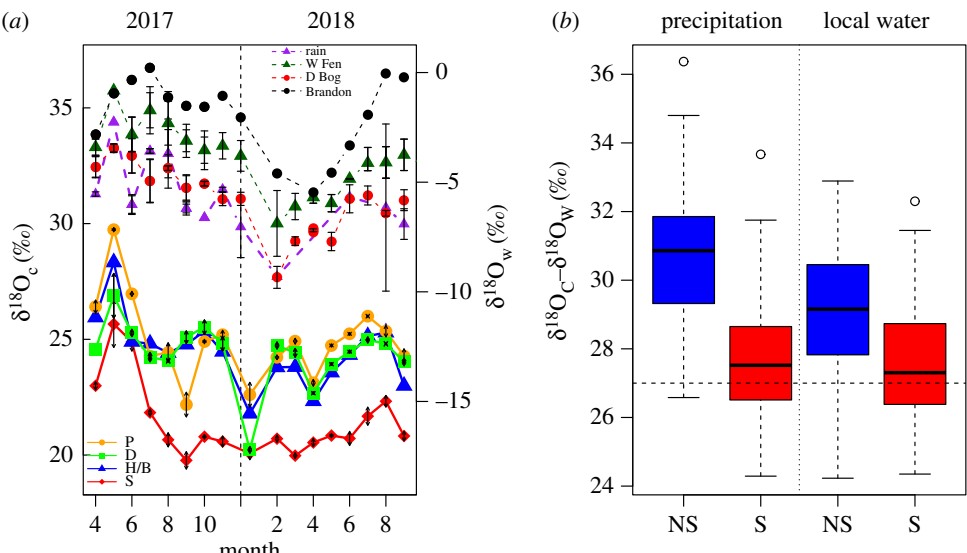

**Figure 2.** (a) Mean (±s.e.) $\delta^{18}O$ values over time for cellulose (S: Sphagnales (red), H/B: Hypnales/Bryales (blue), D: Dicranales (green), and P: Polytrichales (orange) solid lines, left-hand axis), and fresh water samples (rain (purple), W Fen: Wicken Fen (dark green), D Bog: Dersingham Bog (red), Brandon (black)), right-hand axis. (b) Boxplot of the offset between isotopic composition of cellulose and contemporary precipitation and local water for non-Sphagnum (blue, $n = 131$) and Sphagnum (red, $n = 72$) mosses. (Online version in colour.)

sustained a more consistent $CO_2$ diffusive supply and sustained photosynthetic activity, as compared to Hypnale and Bryale mosses. The *Polytrichale* species sampled (*P. commune* for the valley mire at Dersingham Bog; *P. formosum* for the dry heathland at Brandon) acted as climatic indicators within and between such contrasting habitats.

Preserved organic matter in peat-cores provide palaeoclimate records which integrate the relatively stable environmental conditions supporting sufficient net carbon assimilation for growth. The poikilohydric physiology of mosses balances diffusion limitation imposed by external films of water, relative to maximal carbon assimilation as water evaporates. *Sphagnum* species, and Polytrichales, by maintaining a more constant tissue water content and photosynthetic carbon gain, do not go through the high frequency desiccation-rehydration cycles of the Dicranales, Hypnale, and Bryales, but will be affected on a seasonal basis by changes in water table depth and freeze–thaw cycling, thus *Sphagnum* and Polytrichale residual organic matter act as better climatic indicators over an extended growing season.

## (b) Chlorophyll fluorescence indicates photosynthetic limitation imposed by drought or freezing conditions

The capacity for the Polytrichales and Sphagnales to sustain photosynthesis was illustrated by high electron transport rates (ETR, a proxy for $CO_2$ uptake), even on bright sunny days when other moss life forms were desiccated (for example, Dicranales: figure 1c). Despite photorespiration potentially limiting assimilation under high light conditions, Polytrichales' capacity to transport water, and the inherent water retention capacity of *Sphagnum*, sustains photosynthesis and organic carbon accumulation during dry, sunny periods.

The Dicranale and Polytrichale samples are from the same families as the Antarctic moss bank species, *Chorisodontium aciphullum* (Dicranale) and *P. commune* (Polytrichale) [4,33]. Despite the potential for variation in the measurements due

to dependence on instantaneous insolation and the non-planar structure of mosses, the ranges of ETR values (figure 2c) are similar to those measured in Antarctica for *C. aciphullum* and the Polytrichale *Polytrichastrum alpinum* [34]. Our ETR data suggest that the Polytrichales will be photosynthetically active for longer than the Dicranales during desiccating drought or freezing conditions, thus in a palaeoclimate record the Polytrichales may reflect a longer growth period through a broader range of conditions. The periods over which Dicranales organic matter is accumulated will represent a narrower range of growing conditions, perhaps with some contribution from more hydrated tissues slightly deeper within the moss bank [35].

Mosses with the very lowest water contents were unable to restore fluorescence yield after a period of dark adaptation (figure 1e), an indication of severely limited carbon assimilation or growth. The continuous presence of sucrose, an osmoticum, in the field-collected non-*Sphagnum* mosses is an important preparation for desiccation (figure 2f; [36]). By contrast, sucrose concentrations are at least 10 times lower in the less desiccation-tolerant *Sphagnum* species. This may be a further explanation for why in desiccation-tolerant mosses (e.g. *S. ruralis*) the $\delta^{13}C$ and $\delta^{18}O$ isotopic compositions reflect different time points [10] as the sucrose is made during drying periods and then converted to cellulose during the subsequent wet period. Conversely, in the *Sphagnum* spp., which grow in wet sites and thus have a perpetually high water content, little sucrose is retained for osmotic protection so assimilation into sucrose and then cellulose formation are concurrent and the isotope signals correlated [1].

## (c) Stable isotope signals in precipitation and tissue water integrate seasonal inputs and evaporative demand

Precipitation was likely to be derived from similar moisture sources and precipitation events across the geographically close field sites. Indeed, at the perpetually wet Dersingham

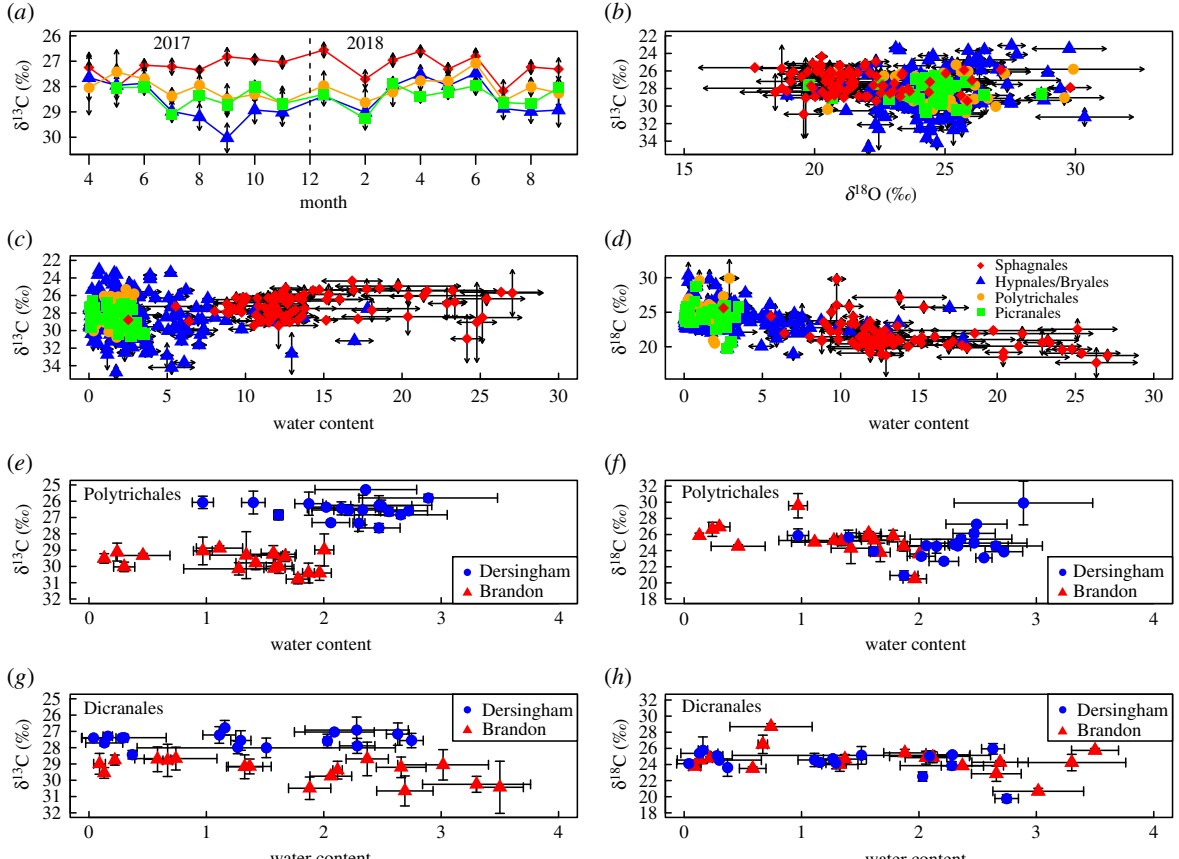

**Figure 3.** Stable isotope composition of growing tip cellulose of *Sphagnum* (red diamonds), *Polytrichale* (orange circles), Hypnales and Bryales (blue triangles), and Dicranales (green squares) mosses: (*a*) monthly mean $\delta^{13}C_C$ (±s.e.); (*b*) $\delta^{18}O_C$ as a function of $\delta^{13}C_C$; mean (*c*) $\delta^{13}C_C$ and (*d*) $\delta^{18}O_C$ as a function of relative water content for each moss species at each site (±s.e., $n = 3$); significant linear model (solid line) plotted through all samples in (*c*). Mean stable isotope composition of comparable mosses from different locations as a function of relative water content: (*e*) and (*f*) Polytrichales: *Polytrichum commune* from Dersingham (blue circles) and *Polytrichastrum formosum* from Brandon (red triangles); (*g*) and (*h*) *Dicranum scoparium* from Dersingham (blue circles) and Brandon (red triangles). (Online version in colour.)

Bog, the bog surface water was very similar in isotopic composition to the precipitation (figure 2) and *Sphagnum* $\delta^{18}O_c$ had the most consistent offset from precipitation than any of the other mosses (figure 2*b*). This offset was close to the expected 27‰ synthesis enrichment value [37,38], suggesting water similar in composition to both precipitation and the bog water was present in the leaf during cellulose synthesis, which would be likely in the low level, humid environment of a bog surface. By contrast, for the more desiccation-tolerant mosses, growing in drier, more exposed locations, the relationship with the local water was less direct (figure 2*b*). It is likely that through the more frequent desiccation cycles, water in the moss tissue used for cellulose synthesis underwent evaporative enrichment, above that of the local water, which was already enriched compared to precipitation (figure 2*b*).

### (d) Carbon stable isotopes integrate carbon accumulation for mosses depending on diffusion limitation and rates of desiccation

Photosynthetic carbon accumulation is directly related to organic $\delta^{13}C_c$ values in mosses, mediated by the extent of surface liquid water which impedes the diffusive supply of $CO_2$ [9,10] and regulates the discrimination expressed by the primary carboxylase, Rubisco [39]. For the *Sphagnum* spp. the relatively constant $\delta^{13}C_c$ values tended to lack seasonal

variation (figure 2*a*) and were found across a wide range of tissue water contents (figure 3*c*), although some more depleted signals were less consistent with diffusion limitation. However, at the ecosystem level seasonal changes in biochemical capacity could influence $\delta^{13}C_c$, and a significant input of respiratory $CO_2$, perhaps dependent on water table depth and freeze–thaw cycle dynamics, could alter source $CO_2$ inputs resulting in more negative values of $\delta^{13}C_c$.

The different *Polytrichum* species also demonstrated $\delta^{13}C_c$ values consistent with habitat preferences: although both had a similar range of water content values at the time of sampling, the less enriched $\delta^{13}C_c$ values at the heathland Brandon site, as compared to Dersingham Bog, (figure 3*e*) showed that diffusion limitation was lower during periods of maximum assimilation [9,10]. The higher evaporative demand at the heathland site would increase evapotranspiration but allowed higher discrimination against $^{13}C$ (more negative $\delta^{13}C_c$) when photosynthesis was active. With no species effect, the *Dicranum scoparum* samples from Dersingham and drier Brandon site again had similar water contents during sampling but can be separated by $\delta^{13}C_c$ composition (figure 3*g*) with the $\delta^{13}C_c$ values suggesting that the growth at Brandon occurs under higher evaporative conditions with less diffusional resistance. The sensitivity of both the Polytrichales and the Dicranales growing in contrasting environments to local evaporative conditions is an important indicator for any palaeoclimate conclusions regarding optimal photosynthetic conditions.

## (e) Stable isotopes ($\delta^{18}$O in *Sphagnum* and $\delta^{13}$C in *Polytrichum*) provide complementary organic material markers for peat bank climatic inputs

A critical question arising from our earlier work on the interpretation of palaeoclimate archives [4,40–42] has been how to interpret isotopic signals ($\delta^{13}C_c$, $\delta^{18}O_c$) across both spatial and temporal scales. The challenge was framed by the possible contributions from changing environmental conditions, length of growing season, degree of moss bank exposure, and interactions between morphology, physiology, and water use by individual species [4]. We identified an urgent need to clarify the climatic impacts on moss life form and habitat in terms of water use and carbon sequestration, whether for the interpretation of past palaeoclimatic conditions or future progressive climate change [4]. By using contrasting habitats within East Anglia, UK, acid mire and alkaline fen, together with nearby heathlands, where mosses varied in the degree of ground flora dominance, we have been able to identify key functional groups, and determinants of their isotopic composition, which dominate current northern boreal and austral peat formations.

Over the two growing seasons the $\delta^{18}O_c$ signal in Sphagnum reflects the seasonal variation in the precipitation for a temperate climate: more isotopically depleted in winter and more enriched in summer. Such seasonal pattern is less apparent in the $\delta^{13}C_c$ signal, which provides an integrated measure reflecting the relatively constant conditions for carbon assimilation and sequestration, and source $CO_2$ inputs, in terms of tissue hydration [41]. By contrast, the tight coupling of the Polytrichale and Dicranale $\delta^{13}C_c$ signals throughout the sampling period suggests that they act as palaeoclimate recorders of optimal assimilatory conditions, while adjacent *Sphagnum* spp. $\delta^{18}O_c$ signals reflect variations in precipitation inputs.

## 5. Conclusion

From a ground-truthing perspective, the prospect of sampling organic material as cellulose from current growth, or even across successive growing seasons within a given strand of moss, can now be used to validate changing climatic inputs. For *Sphagnum* species, which dominate northern boreal ecosystems, the highly hydrated tissues capture and hold precipitation inputs, and transfer the signal into $\delta^{18}O_C$ to represent a marker for changing frequency and intensity of precipitation events across maritime relative to continental landscapes [43–45]. To complement this information, the $\delta^{13}C_c$ signals in associated Polytrichales mosses represent the extent of photosynthetic optimality, which should in turn be able to ground truth remote fluorescence signals across the entire community.

From the stable isotope values measured in real-time on a seasonal basis across East Anglia, UK, the outputs are consistent with work in high latitude moss banks. Here, contrasting signals from contrasting functional groups (Sphagnales, Dicranales, Polytrichales), representing both polar and temperate habitats, allow functional attributes to be partitioned according to $\delta^{18}O$ or $\delta^{13}C$ responses, on the basis of morphological and physiological adaptation to water supply and evaporative demand.

Data accessibility. Research data associated with this article can be accessed through the NERC Environmental Information Data Centre at https://doi.org/10.5285/249034d3-2f4d-42c6-a3cd-113ff3a960c5 [46] and https://doi.org/10.5285/1e57032c-327a-49c7-bb2e-43921dc116e2 [47]. The data are provided in electronic supplementary material [48].

Authors' contributions. J.R.: conceptualization, data curation, formal analysis, funding acquisition, investigation, methodology, project administration, resources, supervision, writing—original draft, writing—review, and editing; S.Y.: formal analysis, investigation, methodology, writing—review and editing; H.G.: conceptualization, funding acquisition, methodology, project administration, resources, supervision, writing—review and editing. All authors gave final approval for publication and agreed to be held accountable for the work performed therein.

Competing interests. We declare we have no competing interests.

Funding. This work was supported by NERC (grant no. RG74445).

Acknowledgements. The authors would like to thank the Natural England (Ash Murray and Fiona Hinds) for access to Dersingham Bog, the National Trust for access to Wicken Fen, the Brandon Country Park for access to sample there, and James Rolfe (Godwin Laboratory) and David Hughes (Lancaster Environment Centre) for assistance with isotope analysis.

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
