## [Peer Review File · Proceedings of the Royal Society B: Biological Sciences]

Review History

RSPB-2021-1194.R0 (Original submission)

Review form: Reviewer 1

Recommendation

Major revision is needed (please make suggestions in comments)

Scientific importance: Is the manuscript an original and important contribution to its field?

Good

General interest: Is the paper of sufficient general interest?

Acceptable

Quality of the paper: Is the overall quality of the paper suitable?

Good

Is the length of the paper justified?

Yes

Should the paper be seen by a specialist statistical reviewer?

No

Do you have any concerns about statistical analyses in this paper? If so, please specify them explicitly in your report.

No

It is a condition of publication that authors make their supporting data, code and materials available - either as supplementary material or hosted in an external repository. Please rate, if applicable, the supporting data on the following criteria.

Is it accessible?

Yes

Is it clear?

N/A

Is it adequate?

N/A

Do you have any ethical concerns with this paper?

No

Comments to the Author

RSPB Manuscript Number: RSPB-2021-1194

General comments:

The authors present data, collected in the field, on temporal variation in moss chlorophyll fluorescence activity (as a proxy for photosynthetic capacity changes), moss water content and associated changes to the carbon and oxygen isotope composition of several contrasting moss species. In addition, the authors compare measurements of the stable carbon and oxygen isotope composition of the contemporary moss samples they studied with herbarium specimens of the same species collected at earlier times (with herbarium samples primarily collected near 1960).

The data presented illustrate that the dominant environmental controls on the stable isotope composition of the mosses differ among contrasting moss functional types. For example, the oxygen isotope composition of Sphagnum moss species appears to be primarily controlled by seasonal variation in the oxygen isotope composition of precipitation inputs, and the carbon isotope composition of Poltrichum groups are strongly influenced by differences in site water availability and water content effects, which can influence carbon dioxide supply to the photosynthetic tissues and changes in moss photosynthetic capacity.

The datasets associated with this manuscript are interesting and useful for helping to interpret moss stable isotope measurements collected from peat cores and subsequently analyzed to provide insights about past climatic changes, as the authors promote in the manuscript text.

While I am generally supportive of the manuscript, I do have some criticisms about the some of the interpretations, which I find gloss over some important (at least to me) complications and subtleties in factors that influence the carbon isotope composition of moss during photosynthetic gas exchange. In addition, it is my opinion that the quality of the figures could (and should) be improved, which would make it easier to understand the main messages the authors are trying to convey.

The C-13 signal in organic tissues of mosses is the result of assimilation integrated over the life of the plant. It will, therefore, be a function of the short-term discrimination weighted by the respective assimilation rates and the length of time those assimilation rates occurred throughout the life of the plant. During periods when the moss is relatively dry, they would be expected to have large C-13 discrimination values because of low diffusional limitation to source carbon

dioxide, but they will also have lower rates of carbon dioxide assimilation. In addition, when comparing data in Figures 3e and 3g for species of Polytrichales and Dicranales at sites with contrasting water availability, the authors only focus on water availability, but the data show that moss water content can be the same at the contrasting sites (or even higher at the dry site), which likely implies that the relationship between moss water content and C-13 discrimination is also different between the species of Polytrichales and Dicranales that are present at the sites with different water availability. As noted in this manuscript and other publications, Sphagnum tend to have very high water contents, this can result in Sphagnum moss being frozen (incased in ice) early in the Spring and also in Autumn after freezing temperatures occur in boreal environments. This is associated with significant seasonal changes in the Rubisco Vcmax values of Sphagnum moss, an indicator of significant seasonal changes in the biochemical capacity for photosynthesis. As a result, seasonal changes in Sphagnum C-13 discrimination and changes in tissue C-13 isotope composition are affected by factors other than just seasonal changes in moss water content. Other moss types, that have lower water contents, are not as susceptible to these strong seasonal freeze-thaw effects on photosynthetic activity. Sphagnum water contents in many boreal peatlands are also strongly influenced by the changes in the water table depth in the peatland, and so are not always strongly affected by short-term changes in precipitation inputs (although precipitation inputs can also influence the water table depth – albeit with lag effects). Issues raised in this paragraph illustrate some of the complications I indicated, and these issues are largely ignored by discussions and interpretation presented in this manuscript. The authors should make some attempt to revise the manuscript to address these criticisms and complications in factors that influence photosynthetic discrimination for carbon and oxygen stable isotopes.

In my opinion there are several aspects of the data presentation in the Figures that could be improved, as described below.

First, the x-axis (Time (month)) for parts of Figures 1-3, would be improved if the scale for the separate years is separated and presented in increments of 2 from 0 to 12. The years (2017 and 2018) could be shown on the top (outside) side of the relevant boxes. This change would make it easier to immediately identify the winter and summer month time periods that were sampled in the two study years.

Second, the axis (x and y axes) labels for moss water content should be listed as “Water Content (RWC)” in Figures 1 and 3.

Third, the data in Figure 1f should be presented as a box plot that compares data for Sphagnum versus the other moss types (non-Sphagnum) – as this is the purpose of the figure as discussed in the text (Lines 137-138).

Fourth, for Figure 2a (which has too much data and it is difficult to see the patterns in the data), the data should be separated, with Sphagnum and non-Sphagnum data pu into separate boxes or separate components of a multi-box figure. The data currently plotted in Figure 2b should be presented as a box plot that compares (median and other associated variation within the data set) for the “cellulose water offset” in the Sphagnum versus the Non-Sphagnum data sets. The temporal variation currently presented in Figure 2b is not really necessary to the point of the comparison being made, and a box plot graph would better summarize the data set for the purposes it is presented in the text of the manuscript. It should also be noted in relation to the data for “rain” data presented in these graphs that only the stable isotope composition is presented without consideration of variation in rainfall amounts – which could be a complicating factor that the authors should comment on. The manuscript text (Lines 140-142) that describes the O-18 composition of rain is weak, because it does not explain the time frame for the measurements very well. In addition, there is no information presented about the distance from where the rain samples were collected – in comparison to the where the moss samples were collected, so the reader cannot fully appreciate whether a fair comparison is being made between the isotope composition of precipitation inputs and the isotope composition of moss at different

field sites.

Fifth, since data for the herbarium specimens presented in Figure 4 almost exclusively represent samples collected in about 1960 (one Sphagnum sample excepted), I find it awkward to present the data using time on the x-axis. I suggest that a box plot showing a comparison of the isotope compositions measured during the 2017-18 sampling versus herbarium samples collected in the 1960s (for a given taxonomic group) would make a more appropriate figure and better match the manner that the data are presented in the Results (Lines 173-180) and Discussion sections.

Other specific comments:

Line 198: delete "the vitality of"

Line 243 (and elsewhere): I find "desiccation-avoiding" an awkward descriptor applied Sphagnum mosses, as such a statement can be made about all plants and Sphagnum is normally only present in sites that are very wet (with a high water table). Some better description should be considered for Sphagnum mosses.

Title: I find the title somewhat misleading, as data presented in this manuscript does not directly address past climatic signals. The paper more directly addresses the following topic: Seasonal measurements of moss physiology and differences between moss functional types in the environmental controls on their carbon and oxygen stable isotope composition. This text is quite long for a title – but the authors could work to develop a new (shorter) title along these lines.

Review form: Reviewer 2

Recommendation

Major revision is needed (please make suggestions in comments)

Scientific importance: Is the manuscript an original and important contribution to its field?

Excellent

General interest: Is the paper of sufficient general interest?

Good

Quality of the paper: Is the overall quality of the paper suitable?

Acceptable

Is the length of the paper justified?

Yes

Should the paper be seen by a specialist statistical reviewer?

No

Do you have any concerns about statistical analyses in this paper? If so, please specify them explicitly in your report.

Yes

It is a condition of publication that authors make their supporting data, code and materials available - either as supplementary material or hosted in an external repository. Please rate, if applicable, the supporting data on the following criteria.

Is it accessible?

Yes

Is it clear?

Yes

Is it adequate?

Yes

Do you have any ethical concerns with this paper?

No

Comments to the Author

Royles et al have conducted a very detailed and ambitious study tracking the $^{13}\text{C}/^{18}\text{O}$ isotopic composition over time in several species. Despite much research we still have little high resolution data on isotopic composition in mosses and a time series like this is rather unique. Interpreting moss $^{13}\text{C}/^{18}\text{O}$ data is challenging as there is many factors at play. I think this is also shown in this study. I really like the authors explanations of how the isotopic composition can change in mosses (Introduction), I have not seen it explained so succinctly and understandable before. This study has potential but the current version feels a bit unpolished in many ways. I have divided up my comments into Major and minor. Hope they will help the authors.

Major comments

Abstract

L24-26. These things that were not studied (remote sensing ChlF1) and it is not clear from the study how ChlF1 and isotopic monitoring can actually help with this.

Introduction

The Aims/Questions are not clear. It was first in the Discussion I understood why this study was conducted, and why sampling was done in a certain way. The introduction have a really good section on the controlling factors of $^{13}\text{C}/^{18}\text{O}$ but I miss a link to the aims of this study and why many response variables were measured (eg sucrose). It is also not clear to me why herbarium specimens are included. To be honest, I dont see such $^{13}\text{C}/^{18}\text{O}$ data relevant at all in this study but I may have missed something. The authors need to explain this, or remove that part.

Methods

A lot of basic information is missing here. Just to mention some: The overall study design, sampling frequency in the field at the different sites, how was fresh water sampled in the field?, why were the loggers in the common garden and how relevant is that information?, what part of the moss was sampled (how many shoots?)?, how were the statistical analyses done? Regarding the statistics on ^{18}O vs RWC (which is a key graph): The data is not independent here (repeated measurement, different species) and I wonder if this was considered in the analysis?

Results/Discussion

I really like the data on sucrose, but I would like to see how it changed over time as well (assuming it was measured at each time point - it was not clear from the Methods).

L160 and general on the conclusions of the paper:

RWC was measured at the time of tissue collection: But incorporation of C and O must have happened before sampling, ie conditions before sampling (you take a big tissue sample representing a longer time period). But there appears to be no lag in the data. I found this a bit troubling but maybe there is a good explanation for it. Also, it seems like this study find a signal in ^{18}O also in the winter, when growth is low. Or at least that is my interpretation ($^{18}\text{O}_\text{c}$ - $^{18}\text{O}_\text{water}$ relationship).

Little growth in the winter....still O18 follows rain O18?

L304-L316. I dont think the authors can conclude this. To me this is more of a general discussion.

Minor comments

L124 and Fig 1. Be specific on what ChlF1 parameters that you mean. Yield can be in dark or light, and in the figure use eg Fv/Fm etc on the axes.

L134-135. Is there some stats supporting this "higher proportion"?

L260. Should this be Fig 3d (not 3b)?

Fig 1a. Remove "(DW)" as it is not a unit.

Fig 3d. Different Sphagnum species here, I wonder how they aggregate along this RWC gradient. This may weaken the argument on L260.

Decision letter (RSPB-2021-1194.R0)

28-Jun-2021

Dear Ms Royles:

I am writing to inform you that your manuscript RSPB-2021-1194 entitled "Seasonal measurements of moss physiology and stable isotope composition of marker species identify contrasting past and present climatic signals" has, in its current form, been rejected for publication in Proceedings B.

This action has been taken on the advice of referees, who have recommended that substantial revisions are necessary. With this in mind we would be happy to consider a resubmission, provided the comments of the referees are fully addressed. However please note that this is not a provisional acceptance.

To upload a resubmitted manuscript, log into <http://mc.manuscriptcentral.com/prsb> and enter your Author Centre, where you will find your manuscript title listed under "Manuscripts with

Decisions." Under "Actions," click on "Create a Resubmission." Please be sure to indicate in your cover letter that it is a resubmission, and supply the previous reference number.

Sincerely,
Dr Maurine Neiman
mailto: proceedingsb@royalsociety.org

Associate Editor
Board Member: 1
Comments to Author:

The reviewers both found a lot of value in this manuscript and the connection between moss physiology and stable isotope composition is very clearly drawn. The contrast between Sphagnum and the Polytrichales group is also interesting given the very different physiologies and habitats of the two groups of species. Both reviewers found the manuscript interesting but also had many points for improvement.

One of the most important parts to improve is the incomplete integration of the past (herbarium) and present parts of the study. The importance of this integration is reflected in the title "identify contrasting past and present climatic signals". But, strangely, the past part of the study is not mentioned in the abstract and what the title actually means is not clear to me. There is a big gap in the integration of the analyses as well.

Reviewer 2 is also puzzled by this: "I miss a link to the aims of this study and why many response variables were measured (eg sucrose). It is also not clear to me why herbarium specimens are included." Reviewer 1 is also confused by the presentation of the historical data (figure 4).

All this implies that a major restructure and more integration of the analysis is necessary. My suggestion is to rearrange the manuscript around this exciting comment from the discussion:

"The tight coupling of the Polytrichale and Dicranale $\delta^{13}\text{C}$ signals throughout the sampling period, suggests that they both have the potential to act as palaeoclimate recorders of assimilatory conditions, whilst the Sphagnum spp. reflects precipitation inputs. Herbariums provide a largely untapped resource of moss organic material from the past that complements peat depth profiles."

This is a clear, exciting, and testable hypothesis. Furthermore, it seems as if the authors seem to have the data in hand to test it--or in any case additional data is relatively easy to obtain. (Note this assumes that past climate data should be obtainable for the time/date of the collections, which I think is a reasonable assumption.)

If that comment can be moved to the aims, and then a clear test of that hypothesis performed with the herbarium data, then I believe that will resolve the confusion that both reviewers and I felt reading this manuscript. This is neither a small restructure nor a simple additional analysis, but this should make for much more clarity and a vastly improved manuscript. It is also not clear what the results of the hypothesis test would be--but I believe either result would be interesting, given the comparison with the contemporary data.

The other issue that is addressed verbally (but not quantitatively) with regard to this is the changing atmospheric ^{13}C through time. Because the atmospheric trend in ^{13}C is well-documented, both present and past data could be pretty simply adjusted from little-delta to big-delta and then allow for an apples-to-apples comparison between the contemporary and the herbarium data.

Reviewer(s)¹ Comments to Author:

Referee: 1

Comments to the Author(s)

RSPB Manuscript Number: RSPB-2021-1194

General comments:

The authors present data, collected in the field, on temporal variation in moss chlorophyll fluorescence activity (as a proxy for photosynthetic capacity changes), moss water content and associated changes to the carbon and oxygen isotope composition of several contrasting moss species. In addition, the authors compare measurements of the stable carbon and oxygen isotope composition of the contemporary moss samples they studied with herbarium specimens of the same species collected at earlier times (with herbarium samples primarily collected near 1960).

The data presented illustrate that the dominant environmental controls on the stable isotope composition of the mosses differ among contrasting moss functional types. For example, the oxygen isotope composition of Sphagnum moss species appears to be primarily controlled by seasonal variation in the oxygen isotope composition of precipitation inputs, and the carbon isotope composition of Poltrichum groups are strongly influenced by differences in site water availability and water content effects, which can influence carbon dioxide supply to the photosynthetic tissues and changes in moss photosynthetic capacity.

The datasets associated with this manuscript are interesting and useful for helping to interpret moss stable isotope measurements collected from peat cores and subsequently analyzed to provide insights about past climatic changes, as the authors promote in the manuscript text.

While I am generally supportive of the manuscript, I do have some criticisms about some of the interpretations, which I find gloss over some important (at least to me) complications and subtleties in factors that influence the carbon isotope composition of moss during photosynthetic gas exchange. In addition, it is my opinion that the quality of the figures could (and should) be improved, which would make it easier to understand the main messages the authors are trying to convey.

The C-13 signal in organic tissues of mosses is the result of assimilation integrated over the life of the plant. It will, therefore, be a function of the short-term discrimination weighted by the respective assimilation rates and the length of time those assimilation rates occurred throughout the life of the plant. During periods when the moss is relatively dry, they would be expected to have large C-13 discrimination values because of low diffusional limitation to source carbon dioxide, but they will also have lower rates of carbon dioxide assimilation. In addition, when comparing data in Figures 3e and 3g for species of Polytrichales and Dicranales at sites with contrasting water availability, the authors only focus on water availability, but the data show that moss water content can be the same at the contrasting sites (or even higher at the dry site), which likely implies that the relationship between moss water content and C-13 discrimination is also different between the species of Polytrichales and Dicranales that are present at the sites with different water availability. As noted in this manuscript and other publications, Sphagnum tend to have very high water contents, this can result in Sphagnum moss being frozen (incased in ice) early in the Spring and also in Autumn after freezing temperatures occur in boreal environments. This is associated with significant seasonal changes in the Rubisco V_{max} values of Sphagnum moss, an indicator of significant seasonal changes in the biochemical capacity for photosynthesis. As a result, seasonal changes in Sphagnum C-13 discrimination and changes in tissue C-13 isotope composition are affected by factors other than just seasonal changes in moss water content. Other moss types, that have lower water contents, are not as susceptible to these strong seasonal freeze-thaw effects on photosynthetic activity. Sphagnum water contents in many boreal peatlands are also strongly influenced by the changes in the water table depth in the peatland, and so are not always strongly affected by short-term changes in precipitation inputs (although precipitation inputs can also influence the water table depth – albeit with lag effects). Issues

raised in this paragraph illustrate some of the complications I indicated, and these issues are largely ignored by discussions and interpretation presented in this manuscript. The authors should make some attempt to revise the manuscript to address these criticisms and complications in factors that influence photosynthetic discrimination for carbon and oxygen stable isotopes.

In my opinion there are several aspects of the data presentation in the Figures that could be improved, as described below.

First, the x-axis (Time (month)) for parts of Figures 1-3, would be improved if the scale for the separate years is separated and presented in increments of 2 from 0 to 12. The years (2017 and 2018) could be shown on the top (outside) side of the relevant boxes. This change would make it easier to immediately identify the winter and summer month time periods that were sampled in the two study years.

Second, the axis (x and y axes) labels for moss water content should be listed as "Water Content (RWC)" in Figures 1 and 3.

Third, the data in Figure 1f should be presented as a box plot that compares data for Sphagnum versus the other moss types (non-Sphagnum) – as this is the purpose of the figure as discussed in the text (Lines 137-138).

Fourth, for Figure 2a (which has too much data and it is difficult to see the patterns in the data), the data should be separated, with Sphagnum and non-Sphagnum data put into separate boxes or separate components of a multi-box figure. The data currently plotted in Figure 2b should be presented as a box plot that compares (median and other associated variation within the data set) for the "cellulose water offset" in the Sphagnum versus the Non-Sphagnum data sets. The temporal variation currently presented in Figure 2b is not really necessary to the point of the comparison being made, and a box plot graph would better summarize the data set for the purposes it is presented in the text of the manuscript. It should also be noted in relation to the data for "rain" data presented in these graphs that only the stable isotope composition is presented without consideration of variation in rainfall amounts – which could be a complicating factor that the authors should comment on. The manuscript text (Lines 140-142) that describes the O-18 composition of rain is weak, because it does not explain the time frame for the measurements very well. In addition, there is no information presented about the distance from where the rain samples were collected – in comparison to the where the moss samples were collected, so the reader cannot fully appreciate whether a fair comparison is being made between the isotope composition of precipitation inputs and the isotope composition of moss at different field sites.

Fifth, since data for the herbarium specimens presented in Figure 4 almost exclusively represent samples collected in about 1960 (one Sphagnum sample excepted), I find it awkward to present the data using time on the x-axis. I suggest that a box plot showing a comparison of the isotope compositions measured during the 2017-18 sampling versus herbarium samples collected in the 1960s (for a given taxonomic group) would make a more appropriate figure and better match the manner that the data are presented in the Results (Lines 173-180) and Discussion sections.

Other specific comments:

Line 198: delete "the vitality of"

Line 243 (and elsewhere): I find "desiccation-avoiding" an awkward descriptor applied Sphagnum mosses, as such a statement can be made about all plants and Sphagnum is normally only present in sites that are very wet (with a high water table). Some better description should be considered for Sphagnum mosses.

Title: I find the title somewhat misleading, as data presented in this manuscript does not directly address past climatic signals. The paper more directly addresses the following topic: Seasonal measurements of moss physiology and differences between moss functional types in the environmental controls on their carbon and oxygen stable isotope composition. This text is quite long for a title – but the authors could work to develop a new (shorter) title along these lines.

Referee: 2

Comments to the Author(s)

Royles et al have conducted a very detailed and ambitious study tracking the $^{13}\text{C}/^{18}\text{O}$ isotopic composition over time in several species. Despite much research we still have little high resolution data on isotopic composition in mosses and a time series like this is rather unique. Interpreting moss $^{13}\text{C}/^{18}\text{O}$ data is challenging as there is many factors at play. I think this is also shown in this study. I really like the authors explanations of how the isotopic composition can change in mosses (Introduction), I have not seen it explained so succinctly and understandable before. This study has potential but the current version feels a bit unpolished in many ways. I have divided up my comments into Major and minor. Hope they will help the authors.

Major comments

Abstract

L24-26. These things that were not studied (remote sensing ChlFl) and it is not clear from the study how ChlFl and isotopic monitoring can actually help with this.

Introduction

The Aims/Questions are not clear. It was first in the Discussion I understood why this study was conducted, and why sampling was done in a certain way. The introduction have a really good section on the controlling factors of $^{13}\text{C}/^{18}\text{O}$ but I miss a link to the aims of this study and why many response variables were measured (eg sucrose). It is also not clear to me why herbarium specimens are included. To be honest, I dont see such $^{13}\text{C}/^{18}\text{O}$ data relevant at all in this study but I may have missed something. The authors need to explain this, or remove that part.

Methods

A lot of basic information is missing here. Just to mention some: The overall study design, sampling frequency in the field at the different sites, how was fresh water sampled in the field?, why were the loggers in the common garden and how relevant is that information?, what part of the moss was sampled (how many shoots?)?, how were the statistical analyses done? Regarding the statistics on ^{18}O vs RWC (which is a key graph): The data is not independent here (repeated measurement, different species) and I wonder if this was considered in the analysis?

Results/Discussion

I really like the data on sucrose, but I would like to see how it changed over time as well (assuming it was measured at each time point – it was not clear from the Methods).

L160 and general on the conclusions of the paper:

RWC was measured at the time of tissue collection: But incorporation of C and O must have happened before sampling, ie conditions before sampling (you take a big tissue sample representing a longer time period). But there appears to be no lag in the data. I found this a bit troubling but maybe there is a good explanation for it. Also, it seems like this study find a signal in ^{18}O also in the winter, when growth is low. Or at least that is my interpretation ($^{18}\text{O}_c$ – $^{18}\text{O}_{\text{water}}$ relationship).

Little growth in the winter....still O_{18} follows rain O_{18} ?

L304-L316. I dont think the authors can conclude this. To me this is more of a general discussion.

Minor comments

L124 and Fig 1. Be specific on what ChlF1 parameters that you mean. Yield can be in dark or light, and in the figure use eg Fv/Fm etc on the axes.

L134-135. Is there some stats supporting this "higher proportion"?

L260. Should this be Fig 3d (not 3b)?

Fig 1a. Remove "(DW)" as it is not a unit.

Fig 3d. Different Sphagnum species here, I wonder how they aggregate along this RWC gradient. This may weaken the argument on L260.

Author's Response to Decision Letter for (RSPB-2021-1194.R0)

See Appendices A & B.

RSPB-2021-2470.R0

Review form: Reviewer 1

Recommendation

Accept with minor revision (please list in comments)

Scientific importance: Is the manuscript an original and important contribution to its field?

Good

General interest: Is the paper of sufficient general interest?

Good

Quality of the paper: Is the overall quality of the paper suitable?

Good

Is the length of the paper justified?

Yes

Should the paper be seen by a specialist statistical reviewer?

No

Do you have any concerns about statistical analyses in this paper? If so, please specify them explicitly in your report.

No

It is a condition of publication that authors make their supporting data, code and materials available - either as supplementary material or hosted in an external repository. Please rate, if applicable, the supporting data on the following criteria.

Is it accessible?

Yes

Is it clear?

N/A

Is it adequate?

N/A

Do you have any ethical concerns with this paper?

No

Comments to the Author

General comments:

I have read the revised manuscript and the responses provided by the authors to the original reviewers' comments. The revised manuscript has fully addressed my previous criticisms. In my opinion the revised manuscript is much improved over the original submission.

I have noted only some very minor corrections (listed below) that should be made to the final manuscript.

Specific comments:

Line 185: the data listed here (29 per mil for precipitation, and 31 per mil for local water) is not consistent with the data shown in Figure 2b (which has these values reversed).

Line 194: "was" should be "were"

Line 198: I think explicit mention should be made here that it is the delta 18O values that are being referred to. So change text to, "... lower end of the delta 18O range ...".

Lines 200-207: there is inconsistent use of tense. Specifically, I suggest that the following word changes should be made:

Line 203 change "is" to "was"

Line 204: change "are" to "were"

Line 205: change "are" to "were"

Line 254: "mossbank" should be "moss bank"

Line 304: "peatbank" should be "peat bank"

Review form: Reviewer 2

Recommendation

Accept with minor revision (please list in comments)

Scientific importance: Is the manuscript an original and important contribution to its field?

Excellent

General interest: Is the paper of sufficient general interest?

Good

Quality of the paper: Is the overall quality of the paper suitable?

Good

Is the length of the paper justified?

Yes

Should the paper be seen by a specialist statistical reviewer?

No

Do you have any concerns about statistical analyses in this paper? If so, please specify them explicitly in your report.

No

It is a condition of publication that authors make their supporting data, code and materials available - either as supplementary material or hosted in an external repository. Please rate, if applicable, the supporting data on the following criteria.

Is it accessible?

Yes

Is it clear?

Yes

Is it adequate?

Yes

Do you have any ethical concerns with this paper?

No

Comments to the Author

I have previously reviewed this manuscript and I am satisfied with the new revised version. Great work by the authors! A few minor comments:

Statistics: This is very briefly described in the Methods. What is this rmcrr package actually doing? Also, it seems like mixed-effects models are used but why and for what analyses?

L331: I guess this should be 18O and not 13C?

ETR: The authors should remember that ETR is not actual CO₂ uptake but still a proxy. ETR depend on the light conditions and it can be misleading in high light if photorespiration occurs. Some info about the light conditions during ChlFlu measurements would be good to add.

ChlFlu: It isn't that easy to measure ChlFlu on mosses in the field. I still miss information how it was done in practice (eg shoots stick up, how to dark adapt). Or were the ChlFlu measurements done in the lab on the 3 samples (then maybe light conditions were the same every time)?

Thanks for the detailed graph on differences in water content among Sphagnum species. If possible I think such graph could be put in the supplemental (but also for all species).

Author's Response to Decision Letter for (RSPB-2021-2470.R0)

See Appendix C.

Decision letter (RSPB-2021-2470.R1)

08-Dec-2021

Dear Dr Royles

I am pleased to inform you that your manuscript entitled "Stable isotope signals provide seasonal climatic markers for moss functional groups" has been accepted for publication in Proceedings B.

Your article has been estimated as being 8 pages long. Our Production Office will be able to confirm the exact length at proof stage.

Data Accessibility section

Open Access

You are invited to opt for Open Access, making your freely available to all as soon as it is ready for publication under a CCBY licence. Our article processing charge for Open Access is £1700. Corresponding authors from member institutions (<http://royalsocietypublishing.org/site/librarians/allmembers.xhtml>) receive a 25% discount to these charges. For more information please visit <http://royalsocietypublishing.org/open-access>.

Paper charges

Sincerely,

Appendix A

Howard Griffiths,
Professor of Plant Ecology
Fellow of Clare College
Co-Chair, Global Food Security Interdisciplinary Research Centre

Department of Plant Sciences

11th November 2021

Downing Street
Cambridge, CB2 3EA
Telephone: (01223) 333946
Fax: (01223) 333953
E-mail: hg230@cam.ac.uk

Dear Professor Barrett,

We are pleased to re-submit our paper: **Stable isotope signals provide seasonal climatic markers for moss functional groups** for consideration in Proceedings of the Royal Society B (MS ID: RSPB-2021-1194).

High latitude peat accumulations provide a remarkable palaeoclimate archive, whether for Antarctica or for northern boreal forests, and stable isotope markers have the potential to couple precipitation inputs, growth and carbon sequestration activities. However, the relative offsets in $\delta^{13}\text{C}$ and $\delta^{18}\text{O}$ contemporary can vary between contrasting mossbanks, and we have undertaken detailed ground truthing study to compare the dynamics contrasting moss functional groups and associated isotopic markers. Furthermore, mosses make a significant contribution to global carbon cycling in northern regions and are essential to climate modelling. Accordingly, we have also validated moss vitality using chlorophyll fluorescence signals, which could now be interpreted via remote sensing methods.

We show, for contrasting moss orders, that the analysis of stable carbon and oxygen isotopes in the organic matter of can resolve optimality of growth conditions and seasonal variations in source water inputs, respectively. These relationships are maintained over the course of two growing seasons, and three different locations in East Anglia, UK. We appreciated the positive and supportive reviews received for our initial submission, and this resubmitted manuscript is improved in line with their helpful suggestions. We hope the findings will be of interest to your general readership, whether interested in the characterisation of stable isotope biomarkers and palaeoclimatic reconstruction, temperate moss physiology or analysis of current climate change impacts on key areas of carbon sequestration.

Yours,

Howard Griffiths
Professor of Plant Ecology
Tel: PlantSci: 44 (0)1223 (3)33946 Clare College 44 (0)1223 (7)60641

Appendix B

Associate Editor

Board Member: 1

Comments to Author:

The reviewers both found a lot of value in this manuscript and the connection between moss physiology and stable isotope composition is very clearly drawn. The contrast between Sphagnum and the Polytrichales group is also interesting given the very different physiologies and habitats of the two groups of species. Both reviewers found the manuscript interesting but also had many points for improvement.

Thank you for the positive overall summary of the work and the constructive comments from the reviewers and Editor.

One of the most important parts to improve is the incomplete integration of the past (herbarium) and present parts of the study. The importance of this integration is reflected in the title "identify contrasting past and present climatic signals". But, strangely, the past part of the study is not mentioned in the abstract and what the title actually means is not clear to me. There is a big gap in the integration of the analyses as well.

Reviewer 2 is also puzzled by this: "I miss a link to the aims of this study and why many response variables were measured (eg sucrose). It is also not clear to me why herbarium specimens are included." Reviewer 1 is also confused by the presentation of the historical data (figure 4).

All this implies that a major restructure and more integration of the analysis is necessary. My suggestion is to rearrange the manuscript around this exciting comment from the discussion:

"The tight coupling of the Polytrichale and Dicranale $\delta^{13}C_c$ signals throughout the sampling period, suggests that they both have the potential to act as palaeoclimate recorders of assimilatory conditions, whilst the Sphagnum spp. reflects precipitation inputs. Herbariums provide a largely untapped resource of moss organic material from the past that complements peat depth profiles."

This is a clear, exciting, and testable hypothesis. Furthermore, it seems as if the authors seem to have the data in hand to test it--or in any case additional data is relatively easy to obtain. (Note this assumes that past climate data should be obtainable for the time/date of the collections, which I think is a reasonable assumption.)

If that comment can be moved to the aims, and then a clear test of that hypothesis performed with the herbarium data, then I believe that will resolve the confusion that both reviewers and I felt reading this manuscript. This is neither a small restructure nor a simple additional analysis, but this should make for much more clarity and a vastly improved manuscript. It is also not clear what the results of the hypothesis test would be--but I believe either result would be interesting, given the comparison with the contemporary data.

The other issue that is addressed verbally (but not quantitatively) with regard to this is the changing atmospheric ^{13}C through time. Because the atmospheric trend in ^{13}C is well-documented, both present and past data could be pretty simply adjusted from little-delta to big-delta and then allow

for an apples-to-apples comparison between the contemporary and the herbarium data.

Thank you for this helpful summary of the issues that needed clarification to improve the presentation of the paper. We completely agree with this analysis and also the helpful comments of one referee who suggested transposing the opening section of the Discussion to set the scene more clearly at the end of the Introduction. Thus, we have been able to focus more clearly on the key findings, that the oxygen isotopes in Sphagnum provide a clear indicator of changes in local precipitation inputs, whilst the carbon isotopes in sympatric Polytrichale spp provide a marker for the photosynthetic potential for carbon assimilation (and sequestration). However, one casualty of this approach is that we consider the herbarium findings to represent a rather scant and preliminary dataset. The sample size is too small to demonstrate recent climatic variation, and would need a much more detailed explanation of the factors constraining past carbon isotope signals (viz. source CO₂ concentration and isotopic composition). Regretfully we feel that the paper will achieve a higher impact by focussing on the two isotopic signals in contemporary samples, which do offer insights for contrasting seasonal limitations in terms, respectively, of water inputs and photosynthetic potential.

Reviewer(s)' Comments to Author:

Referee: 1

Comments to the Author(s)

RSPB Manuscript Number: RSPB-2021-1194

General comments:

The authors present data, collected in the field, on temporal variation in moss chlorophyll fluorescence activity (as a proxy for photosynthetic capacity changes), moss water content and associated changes to the carbon and oxygen isotope composition of several contrasting moss species. In addition, the authors compare measurements of the stable carbon and oxygen isotope composition of the contemporary moss samples they studied with herbarium specimens of the same species collected at earlier times (with herbarium samples primarily collected near 1960).

The data presented illustrate that the dominant environmental controls on the stable isotope composition of the mosses differ among contrasting moss functional types. For example, the oxygen isotope composition of Sphagnum moss species appears to be primarily controlled by seasonal variation in the oxygen isotope composition of precipitation inputs, and the carbon isotope composition of Poltrichum groups are strongly influenced by differences in site water availability and water content effects, which can influence carbon dioxide supply to the photosynthetic tissues and changes in moss photosynthetic capacity.

The datasets associated with this manuscript are interesting and useful for helping to interpret moss stable isotope measurements collected from peat cores and subsequently analyzed to provide insights about past climatic changes, as the authors promote in the manuscript text.

While I am generally supportive of the manuscript, I do have some criticisms about the some of the interpretations, which I find gloss over some important (at least to me) complications and subtleties

in factors that influence the carbon isotope composition of moss during photosynthetic gas exchange. In addition, it is my opinion that the quality of the figures could (and should) be improved, which would make it easier to understand the main messages the authors are trying to convey.

The figures have been extensively reworked and we hope now provide a clearer depiction of the contrasting responses between individual functional groups

The C-13 signal in organic tissues of mosses is the result of assimilation integrated over the life of the plant. It will, therefore, be a function of the short-term discrimination weighted by the respective assimilation rates and the length of time those assimilation rates occurred throughout the life of the plant. During periods when the moss is relatively dry, they would be expected to have large C-13 discrimination values because of low diffusional limitation to source carbon dioxide, but they will also have lower rates of carbon dioxide assimilation.

The explanation in the introduction is expanded to include this: Line 59: “integrating brief periods of high discrimination and high assimilation rates during dry periods before desiccation and potentially longer periods of lower discrimination and lower assimilation rates during damp periods”

In addition, when comparing data in Figures 3e and 3g for species of Polytrichales and Dicranales at sites with contrasting water availability, the authors only focus on water availability, but the data show that moss water content can be the same at the contrasting sites (or even higher at the dry site), which likely implies that the relationship between moss water content and C-13 discrimination is also different between the species of Polytrichales and Dicranales that are present at the sites with different water availability. As noted in this manuscript and other publications, Sphagnum tend to have very high water contents, this can result in Sphagnum moss being frozen (incased in ice) early in the Spring and also in Autumn after freezing temperatures occur in boreal environments. This is associated with significant seasonal changes in the Rubisco Vcmax values of Sphagnum moss, an indicator of significant seasonal changes in the biochemical capacity for photosynthesis. As a result, seasonal changes in Sphagnum C-13 discrimination and changes in tissue C-13 isotope composition are affected by factors other than just seasonal changes in moss water content. Other moss types, that have lower water contents, are not as susceptible to these strong seasonal freeze-thaw effects on photosynthetic activity. Sphagnum water contents in many boreal peatlands are also strongly influenced by the changes in the water table depth in the peatland, and so are not always strongly affected by short-term changes in precipitation inputs (although precipitation inputs can also influence the water table depth – albeit with lag effects). Issues raised in this paragraph illustrate some of the complications I indicated, and these issues are largely ignored by discussions and interpretation presented in this manuscript. The authors should make some attempt to revise the manuscript to address these criticisms and complications in factors that influence photosynthetic discrimination for carbon and oxygen stable isotopes.

Thank you. These are all highly relevant points, and we have incorporated comments to explain the extent that impact of freezing and desiccation, as well as the close relationship between mire surface water and precipitation inputs. As we have focussed on using the 13C signal in a palaeoclimate context, the signal being interpreted will represent the whole growing season, but be dominated by the times of most optimal conditions – likely to be spring and autumn. Whilst freeze-thaw dynamics play an important role in the physiology of mosses, only a small proportion of growth

will occur during the times that the freeze-thaw dynamics are dominating the system. Similarly, in the height of summer during periods when the water table depth has declined, it is likely that there will be no assimilation in the highly desiccation tolerant mosses, and under these conditions even Sphagnum may well dry and minimise C gain at that point.

Inserted into introduction: “Precipitation rehydrating dry desiccation tolerant moss is likely to undergo evaporative enrichment prior to incorporation into cellulose, whilst in the perpetually wetter environments of *Sphagnum* spp. seasonal changes in water table depth and freeze thaw cycling influence both the water availability and the isotopic composition of the water available during cellulose synthesis “

Inserted into discussion: Whilst at the seasonal and ecosystem level there are other considerations: seasonal changes in biochemical capacity can influence $\delta^{13}\text{C}$, as can any significant input of respiratory CO_2 , water table depth and freeze-thaw cycle dynamics impact how the precipitation mixes with ground water

In my opinion there are several aspects of the data presentation in the Figures that could be improved, as described below.

First, the x-axis (Time (month)) for parts of Figures 1-3, would be improved if the scale for the separate years is separated and presented in increments of 2 from 0 to 12. The years (2017 and 2018) could be shown on the top (outside) side of the relevant boxes. This change would make it easier to immediately identify the winter and summer month time periods that were sampled in the two study years.

Thank you – this is a helpful suggestion and has been changed

Second, the axis (x and y axes) labels for moss water content should be listed as “Water Content (RWC)” in Figures 1 and 3.

This has been changed as suggested

Third, the data in Figure 1f should be presented as a box plot that compares data for Sphagnum versus the other moss types (non-Sphagnum) – as this is the purpose of the figure as discussed in the text (Lines 137-138).

Thank you – this is a helpful suggestion and has been changed

Fourth, for Figure 2a (which has too much data and it is difficult to see the patterns in the data), the data should be separated, with Sphagnum and non-Sphagnum data put into separate boxes or separate components of a multi-box figure. The data currently plotted in Figure 2b should be presented as a box plot that compares (median and other associated variation within the data set) for the “cellulose water offset” in the Sphagnum versus the Non-Sphagnum data sets. The temporal variation currently presented in Figure 2b is not really necessary to the point of the comparison being made, and a box plot graph would better summarize the data set for the purposes it is presented in the text of the manuscript.

The data in figure 2 has been replotted to show the precipitation data, the local water data and the cellulose data . The data has been presented as box plots as suggested with the offsets calculated between the cellulose, local sitting water and monthly precipitation input

It should also be noted in relation to the data for “rain” data presented in these graphs that only the stable isotope composition is presented without consideration of variation in rainfall amounts –

which could be a complicating factor that the authors should comment on. The manuscript text (Lines 140-142) that describes the O-18 composition of rain is weak, because it does not explain the time frame for the measurements very well. In addition, there is no information presented about the distance from where the rain samples were collected – in comparison to the where the moss samples were collected, so the reader cannot fully appreciate whether a fair comparison is being made between the isotope composition of precipitation inputs and the isotope composition of moss at different field sites.

More information has been added into the methods about the sample collection.

“ Fresh water samples (local water) were collected in 10 ml vials from sitting bog water (Dersingham), a lake (Brandon) and a lode (Wicken) at each field site on each visit. Precipitation was collected in a rain gauge near the centre of the field site area (Ely, 52.40° N, 0.26° E, 36 km from Dersingham, 26 km from Brandon, 22 km from Wicken), throughout the sampling period following rain event”

Figure 2a now has the rainfall data plotted alongside the local water collected at each site over time so it is clear to see how the precipitation data is related to the water collected at each site, and indeed is very similar to the surface water at Dersingham bog, the most distant site

Fifth, since data for the herbarium specimens presented in Figure 4 almost exclusively represent samples collected in about 1960 (one Sphagnum sample excepted), I find it awkward to present the data using time on the x-axis. I suggest that a box plot showing a comparison of the isotope compositions measured during the 2017-18 sampling versus herbarium samples collected in the 1960s (for a given taxonomic group) would make a more appropriate figure and better match the manner that the data are presented in the Results (Lines 173-180) and Discussion sections.

As we have explained above, the herbarium data has now been removed from this manuscript

Other specific comments:

Line 198: delete “the vitality of”

Done

Line 243 (and elsewhere): I find “desiccation-avoiding” an awkward descriptor applied Sphagnum mosses, as such a statement can be made about all plants and Sphagnum is normally only present in sites that are very wet (with a high water table). Some better description should be considered for Sphagnum mosses.

.Desiccation-avoider removed – changed to “Sphagnum spp, which grow in wet sites and thus have a perpetually high water content”,

Title: I find the title somewhat misleading, as data presented in this manuscript does not directly address past climatic signals. The paper more directly addresses the following topic: Seasonal measurements of moss physiology and differences between moss functional types in the environmental controls on their carbon and oxygen stable isotope composition. This text is quite long for a title – but the authors could work to develop a new (shorter) title along these lines.

Agreed, new title added: Stable isotope signals provide seasonal climatic markers for moss functional groups

Referee: 2

Comments to the Author(s)

Royles et al have conducted a very detailed and ambitious study tracking the $^{13}\text{C}/^{18}\text{O}$ isotopic composition over time in several species. Despite much research we still have little high resolution data on isotopic composition in mosses and a time series like this is rather unique. Interpreting moss $^{13}\text{C}/^{18}\text{O}$ data is challenging as there is many factors at play. I think this is also shown in this study. I really like the authors explanations of how the isotopic composition can change in mosses (Introduction), I have not seen it explained so succinctly and understandable before. This study has potential but the current version feels a bit unpolished in many ways. I have divided up my comments into Major and minor. Hope they will help the authors.

Thank you for the positive support and helpful comments

Major comments

Abstract

L24-26. These things that were not studied (remote sensing ChlFI) and it is not clear from the study how ChlFI and isotopic monitoring can actually help with this.

Reference to CHIF removed

Introduction

The Aims/Questions are not clear. It was first in the Discussion I understood why this study was conducted, and why sampling was done in a certain way. The introduction have a really good section on the controlling factors of $^{13}\text{C}/^{18}\text{O}$ but I miss a link to the aims of this study and why many response variables were measured (eg sucrose).

We have moved a section from the beginning of the discussion to provide a clear link to the aims of the study.

It is also not clear to me why herbarium specimens are included. To be honest, I dont see such $^{13}\text{C}/^{18}\text{O}$ data relevant at all in this study but I may have missed something. The authors need to explain this, or remove that part.

Herbarium data has been removed as it requires more in detail analsyis that is beyond the current remit to give this analysis the necessary power – see above.

Methods

A lot of basic information is missing here. Just to mention some: The overall study design, sampling frequency in the field at the different sites,

Added: Field work was carried out approximately monthly from April 2017 until September 2018 at three field sites across East Anglia (Dersingham Bog, Brandon Country Park and Wicken Fen; Fig. S1, Table S1) that incorporated bog, heathland, shaded woodland and fen habitats. Target species were identified that represented a range of moss lifeforms, growing under different conditions, which also provided comparable analogues for moss species used in palaeoclimate work. The ectohydric Polytrichales, present in Antarctic peat bank palaeoclimate archives were represented by *Polytrichum commune* (Dersingham) and *Polytrichastrum formosum* (Brandon). *Dicranum scoparium*, a Dicranale and similar to the Antarctic peat bank species *Chorisodontium aciphyllum*, was also present at Dersingham and Brandon. Four *Sphagnum* spp species, characteristic of wet environments were sampled (Dersingham). Finally, four Hypnale (*Pseudoscleropodium purum*, *Pleurozium schreberi*, *Brachythecium rutabulum*, *Calliergonella cuspidate*, *Kindbergia praelonga* and one Bryale species (*Aulacomnium palustre*) were sampled from across the sites

how was fresh water sampled in the field?,

Fresh water samples were collected in 10 ml vials from sitting bog water (Dersingham), a lake (Brandon) and the river (Wicken) at each field site on each visit, Precipitation was collected in a rain gauge near the centre of the field site area (Ely, 52.40° N, 0.26° E, 36 km from Dersingham, 26 km from Brandon, 22 km from Wicken), throughout the sampling period following rain event

why were the loggers in the common garden and how relevant is that information?,

to provide measurements comparable with the meteorological data as installations were not permitted at the sampling locations

what part of the moss was sampled (how many shoots?)?,

Clarified: approximately 4 cm² in area and 2 cm deep, incorporating the growing tips and it was the growing tip that underwent cellulose extraction

how were the statistical analyses done?

2.4 Statistical Analysis

Statistical analysis was carried out in R (v 4.0.5) {R Core Development Team, 2014 #811}. Repeated measures correlation analysis was carried out using the rmcrr package {Bakdash, 2017 #2122}.

Regarding the statistics on 18O vs RWC (which is a key graph): The data is not independent here (repeated measurement, different species) and I wonder if this was considered in the analysis?

The data has been analysed using repeated measures correlation $r=-0.364$, $df=297$, $p<0.001$ rmcrr package {Bakdash, 2017 #2122. The line plotted was an overall mean as the multiple individual species lines generated a very cluttered graph, but this line has been removed for clarity.

Results/Discussion

I really like the data on sucrose, but I would like to see how it changed over time as well (assuming it was measured at each time point – it was not clear from the Methods).

Clarified: A sub-sample of a selection of fresh field mosses harvested in July 2017 underwent enzymatic photo-spectrophotometric sucrose content analysis following standard procedures

L160 and general on the conclusions of the paper:

RWC was measured at the time of tissue collection: But incorporation of C and O must have happened before sampling, ie conditions before sampling (you take a big tissue sample representing a longer time period). But there appears to be no lag in the data. I found this a bit troubling but maybe there is a good explanation for it.

The sampled organic matter analysed for isotopic composition was the top growing tip (a smaller sample than the whole 2 x2x2 cm sample harvested, as has now been clarified in the methods), so will represent recent growth. The measurements of both the cellulose and of the water are integrated over time, presented in Fig 2 as monthly means and both would be expected to change gradually over time. In these temperate mosses it is possible for new growth to occur all year round when conditions are suitable. They won't be growing when frozen / covered in snow, but during warmer winter days they will be rapidly be able to reactivate photosynthesis and assimilate. For the sphagnum mosses, the cellulose value is quite constant throughout the winter months, so it is not possible to define when the growth occurred, but the fluorescence measurements show that the photosynthetic apparatus of the mosses is capable of photosynthesis throughout the year.

L304-L316. I dont think the authors can conclude this. To me this is more of a general discussion.

Removed

Minor comments

L124 and Fig 1. Be specific on what ChlFl parameters that you mean. Yield can be in dark or light, and in the figure use eg Fv/Fm etc on the axes.

This has been altered in the figure axes titles

L134-135. Is there some stats supporting this "higher proportion"?

A higher proportion of low dark yield measurements were found in the Hypnales, Bryales and Dicranale samples than Polytrichales and Sphagnales (χ^2 (1, N=322) = 5.27, p=0.022)

L260. Should this be Fig 3d (not 3b)?

Yes

Fig 1a. Remove "(DW)" as it is not a unit.

Done

Fig 3d. Different Sphagnum species here, I wonder how they aggregate along this RWC gradient. This may weaken the argument on L260.

The species were not plotted separately as it became a very busy graph. Wetter than 20 all the samples except one belong to *Sphagnum fallax*, however *S. fallax* also has samples down to 5 in wetness. Removing the samples >20 the relationship remains significant. With RWC between 5 and 20 the five Sphagnum species are well mixed, with no clear clustering of any of the species

Appendix C

Response to Reviewers: Manuscript ID RSPB-2021-2470

Associate Editor

Comments to Author:

Thanks for your hard work on this revision. The paper does seem much more streamlined and targeted with the new focus and the exclusion of the herbarium data. The reviewers have a number of minor comments to fix. In my view this paper will make a major contribution to the literature on stable isotopes in mosses.

Thank you to the associate editor and the two reviewers for their constructive comments that have really improved our manuscript, we are delighted that it has been accepted for publication.

Reviewer(s)' Comments to Author:

Referee: 1

Comments to the Author(s).

General comments:

I have read the revised manuscript and the responses provided by the authors to the original reviewers' comments. The revised manuscript has fully addressed my previous criticisms. In my opinion the revised manuscript is much improved over the original submission.

Thank you for your support and your help in improving the manuscript with your comments.

I have noted only some very minor corrections (listed below) that should be made to the final manuscript.

Specific comments:

Line 185: the data listed here (29 per mil for precipitation, and 31 per mil for local water) is not consistent with the data shown in Figure 2b (which has these values reversed).

Thank you – the description in the text was reversed and has been corrected

Line 194: “was” should be “were”

Corrected

Line 198: I think explicit mention should be made here that it is the delta 18O values that are being referred to. So change text to, “... lower end of the delta 18O range ...”.

Altered

Lines 200-207: there is inconsistent use of tense. Specifically, I suggest that the following word changes should be made:

Line 203 change "is" to "was" - done

Line 204: change "are" to "were" - done

Line 205: change "are" to "were" - done

Line 254: "mossbank" should be "moss bank" - done

Line 304: "peatbank" should be "peat bank" - done

Referee: 2

Comments to the Author(s).

I have previously reviewed this manuscript and I am satisfied with the new revised version. Great work by the authors! A few minor comments:

Statistics: This is very briefly described in the Methods. What is this rmcrr package actually doing? Also, it seems like mixed-effects models are used but why and for what analyses?

The statistics section has been expanded to include more detail, and also includes references to the packages.

"Statistical analysis was carried out in R (v 4.0.5) (30). The significance of the different proportions of contrasting moss groups with low dark yield measurements was tested using chi-squared test. As they were not normally distributed the difference in sucrose content between *Sphagnum* and non-*Sphagnum* was tested using Wilcoxon signed-rank test. Using the lme4 package (32) mixed-effect model was fitted to the water and $\delta^{18}\text{O}_c$ data with post-hoc pairwise testing using tukey tests and the kenward-rogers degrees of freedom model. The rmcrr package was used to calculate the relationship between relative water content and $\delta^{18}\text{O}_c$ (Fig 3d), taking account of the repeated measurement"

L331: I guess this should be 18O and not 13C? – Yes thanks - corrected

ETR: The authors should remember that ETR is not actual CO₂ uptake but still a proxy. ETR depend on the light conditions and it can be misleading in high light if photorespiration occurs. Some info about the light conditions during ChlFlu measurements would be good to add.

We have not included a specific figure with the light measurements at the time of recording, but the light level is the factor that relates Figure 1b (Fv/Fm) with Figure 1c (ETR) and I have added the equation explicitly to the methods.. At line 162 we state: Partially reflecting degree of exposure and incident light levels, electron transport rate (ETR) was generally highest in the *Sphagnum* mosses, and highest during summer months (Fig 1c).

In the discussion (line 248), we have added the clarification that "ETR, a proxy for CO₂ uptake".

And at line 249: "Despite photorespiration potentially limiting assimilation under high light conditions,"

ChlFlu: It isn't that easy to measure ChlFlu on mosses in the field. I still miss information how it was done in practice (eg shoots stick up, how to dark adapt). Or were the ChlFlu measurements done in the lab on the 3 samples (then maybe light conditions were the same every time)?

We have added information to the methods to clarify this, and how ETR was calculated (from previous comment).

“three measurements (F' , F_m') taken using the leaf clip before harvest on separate sub-samples under ambient light (PAR) which was measured by the light sensor. After harvest, moss samples were stored in sealed plastic bags, and transported to the lab. After at least 30 minutes of dark adaption, three further fluorescence measurements were made in the dark (F , F_m). Fluorescence yield was calculated in the dark $(F_m - F)/F_m$ and light $((F_m' - F')/F_m')$. Electron Transport Rate (ETR) was calculated as: $((F_m' - F')/F_m') \times \text{PAR} \times 0.42$, where PAR is the photosynthetically active radiation and 0.42 is the product of light absorptance by an average green leaf (0.84) times the fraction of absorbed quanta available for photosystem II (0.5).”

Thanks for the detailed graph on differences in water content among Sphagnum species. If possible I think such graph could be put in the supplemental (but also for all species).

This graph has been included in Supplemental (S3)